# Enhanced Latent Space Blind Model for Real Image Denoising via Alternative Optimization

**Chao Ren**[*]    **Yizhong Pan**    **Jie Huang**

College of Electronics and Information Engineering, Sichuan University, Chengdu, China

`chaoren@scu.edu.cn, {panyizhong, huangjiechn}@stu.scu.edu.cn`

## Abstract

Motivated by the achievements in model-based methods and the advances in deep networks, we propose a novel enhanced latent space blind model based deep unfolding network, namely ScaoedNet, for complex real image denoising. It is derived by introducing latent space, noise information, and guidance constraint into the denoising cost function. A self-correction alternative optimization algorithm is proposed to split the novel cost function into three alternative subproblems, *i.e.*, guidance representation (GR), degradation estimation (DE) and reconstruction (RE) subproblems. Finally, we implement the optimization process by a deep unfolding network consisting of GR, DE and RE networks. For higher performance of the DE network, a novel parameter-free noise feature adaptive enhancement (NFAE) layer is proposed. To synchronously and dynamically realize internal-external feature information mining in the RE network, a novel feature multi-modulation attention (FM$^2$A) module is proposed. Our approach thereby leverages the advantages of deep learning, while also benefiting from the principled denoising provided by the classical model-based formulation. To the best of our knowledge, our enhanced latent space blind model, optimization scheme, NFAE and FM$^2$A have not been reported in the previous literature. Experimental results show the promising performance of ScaoedNet on real image denoising. Code is available at `https://github.com/chaoren88/ScaoedNet`.

## 1 Introduction

Image denoising is important for various computer vision tasks, and numerous outstanding methods have been developed [13, 17, 46, 18, 36, 26, 49, 54, 27, 32, 20]. The filtering-based methods are representative, such as the classic local low-pass filtering methods: mean filtering and median filtering. In recent decades, many non-local filtering methods [13, 14, 16, 25] have achieved a great success. However, their results may suffer from blurring artifacts caused by their block-wise operations.

The model-based methods largely depend on image priors, e.g., graph-based [34], sparsity [3, 30, 47], local smoothing [33, 45], and low-rank [17, 44, 46, 43, 55] priors. Pang *et al.* [34] interpreted neighborhood graphs of pixel patches as discrete counterparts of Riemannian manifolds, providing insights into several fundamental aspects of graph Laplacian regularization for denoising. By introducing three weight matrices into the data and regularization terms of the sparse coding framework, Xu *et al.* [47] developed a denoising method to characterise the statistics of real noise and image priors. Gu *et al.* [17] proposed an image restoration method by studying the weighted nuclear norm minimization problem. However, most of these methods focus on removing the additive white Gaussian noise (AWGN), without taking the real noise and external guidance into consideration.

The learning-based methods are promising ways for addressing the image denoising problem. Generally, it can be divided into traditional and deep network-based methods. The deep network-based

---

[*]Corresponding author.

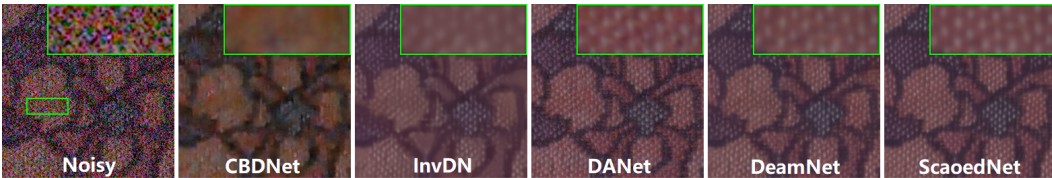

Figure 1: Denoising results on a real noisy image from SIDD dataset [2]. Compared with CBDNet [19], InvDN [27], DANet [49], and DeamNet [36], our ScaoedNet achieves better results.

methods have superiority in denoising due to their excellent modeling ability. For example, the noise map is introduced as the input for removing noise at different levels in [53]. The studies [52, 39, 31] show that it is feasible to learn a single model for blind Gaussian denoising, but these blind models may be over-fitted to AWGN and fail to handle real noise [19]. Different from the previous empirically designed networks, the deep unfolding-based methods [11, 15, 24, 21] were proposed by implementing traditional methods via deep networks. For instance, an advanced image denoising network was developed by solving a fractional optimal control problem in [21].

The above methods generally aim at removing AWGN with regular distributions. However, real image denoising is a more challenging task since the distribution of real noise is more complex. For real image denoising, Guo *et al*. [19] adopted Poisson-Gaussian noise model and presented a blind denoising network. A spatial-adaptive denoising network was proposed in [10] for efficient blind noise removal. Byun *et al*. [7] proposed a fast noise estimation network and an efficient blind-spot network for Poisson-Gaussian noise removal. Anwar *et al*. [5] incorporated feature attention into denoising and developed a one-stage real image denoising network. Cheng *et al*. [12] improved the image denoising performance by plugging the subspace attention module. Liu *et al*. [28] presented an invertible denoising network for real image denoising. Based on the adaptive consistency prior, Ren *et al*. [36] proposed a novel model-based denoising method to inform the design of the network for both synthetic and real denoising. Yue *et al*. [49] proposed a unified framework, namely dual adversarial network, to simultaneously deal with the real noise removal and noise generation tasks.

In this paper, motivated by the advances in deep networks and relying on the rich body of the model-based methods, we propose a novel enhanced latent space (LS) blind model based deep unfolding network for real image denoising. As shown in Fig. 1, the whole architecture of ScaoedNet achieves the best promising visual performance when compared to other denoising methods. The main contributions of our method are as follows:

• For the theoretical novelty, a novel enhanced model-based denoising cost function is proposed based on LS, noise information, and guidance constraint (GC). Then, a self-correction (SC) based alternative optimization algorithm is proposed to split the cost function into three alternative subproblems (*i.e.*, guidance representation (GR), degradation estimation (DE) and reconstruction (RE) subproblems).

• To merge the power of the model-based framework with the recent advances in deep learning for real image denoising, a novel interpretable network (ScaoedNet) is designed via implementing the optimization process by a deep network, which consists of the GR, DE, and RE networks.

• An effective DE network with a novel noise feature adaptive enhancement (NFAE) layer is proposed. NFAE does not add extra parameters to the DE network, and leads to better noise estimation results.

• To enable internal-external feature mining, a novel feature multi-modulation attention residual block (FM$^2$ARB) is proposed in the RE network, which is mainly based on feature self-modulation (FSM) and degradation external-modulation (DEM). FM$^2$ARB can dynamically fuse the encoded degradation representation and the image feature representation, leading to higher performance.

## 2 Proposed Model-based Denoising Method

### 2.1 Enhanced Latent Space Blind Model for Denoising

Let $\mathbf{y} \in \mathbb{R}^{n \cdot c}$ be a noisy image with $c$ channels and $n$ pixels, $\mathbf{x} \in \mathbb{R}^{n \cdot c}$ be its clean version, and $\varphi(\cdot)$ be the regularizer weighted by $\mu$. The traditional model-based denoising method can be given by

$$\hat{\mathbf{x}} = \arg\min_{\mathbf{x}} \mathcal{H}(\mathbf{x}, \mathbf{y}) + \mu\varphi(\mathbf{x}), \tag{1}$$

where $\mathcal{H}(\mathbf{x}, \mathbf{y})$ is the data fidelity which is usually set to $\|\mathbf{y} - \mathbf{x}\|_2^2$. $\varphi(\cdot)$ can be any image regularizers, *e.g.* sparsity prior [3, 30, 47], local smoothing prior [33, 45], and low-rank prior [17, 44, 46, 43], etc.

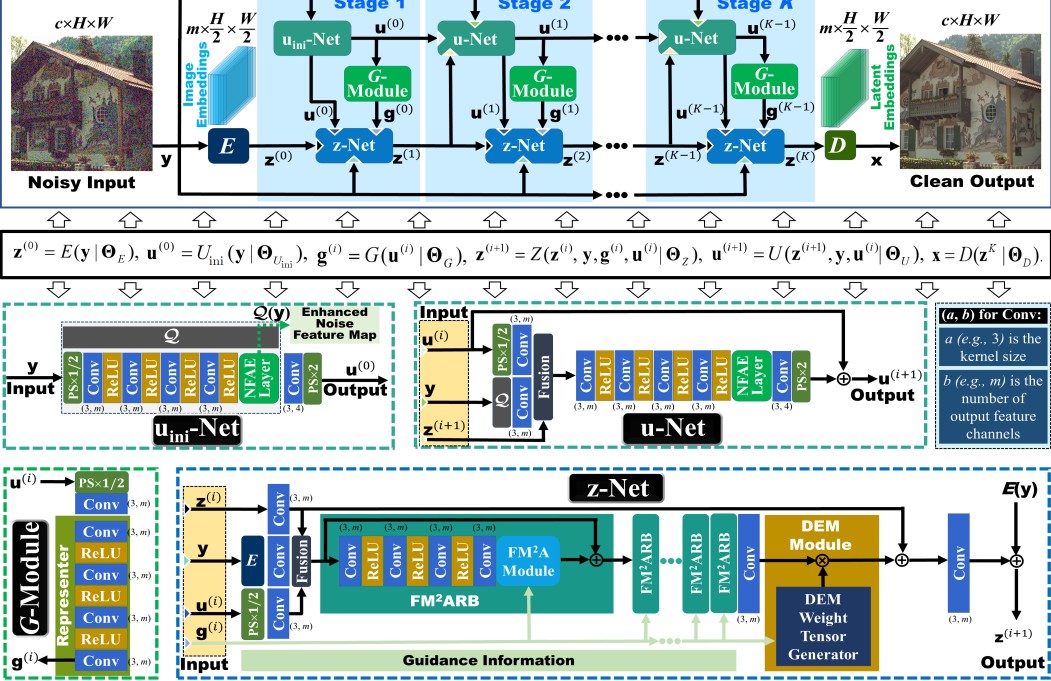

Figure 2: Architecture of ScaoedNet. It alternatively optimizes the guidance representation (GR), degradation estimation (DE) and reconstruction (RE) subproblems in latent space (LS), consisting of LS encoding module ($E$), LS decoding module ($D$), $G$-modules, initial DE network ($\mathbf{u}_{\mathrm{ini}}$-Net), DE networks ($\mathbf{u}$-Nets), and RE networks ($\mathbf{z}$-Nets).

**Analysis and Enhancement.** We perform an in-depth analysis of Eq. (1) in the following:

1) $\mathcal{H}(\mathbf{x}, \mathbf{y})$ in Eq. (1) only utilizes the squared error distance $\|\mathbf{y} - \mathbf{x}\|_2^2$ between $\mathbf{y}$ and $\mathbf{x}$, which is mainly derived from AWGN. Inspired by the latest unfolding works [36, 6], we parameterize the metric $\|\mathbf{y} - \mathbf{x}\|_2^2$ by a pre-specified high-dimensional LS encoding function $E : \mathbb{R}^{n \cdot c} \rightarrow \mathbb{R}^{n \cdot m}$ ($m > c$), to obtain high-dimensional image embeddings $\mathbf{z} = E(\mathbf{x})$ for better performance, instead of limiting the objective to the squared error in low-dimensional image space. It allows us to directly model the image formation process in LS [36, 6], and to integrate LS priors into the prediction.

2) In real image denoising, noise is complex and unknown, and thus a blind model is needed. Since the noise information is largely ignored in Eq. (1), it is difficult to well address the real noise removal problem. By introducing the real noise information and LS, we can learn a more general distance measure $\mathcal{H}(\mathbf{z}, \mathbf{u}, \mathbf{y})$ ($\mathbf{u}$ is the unknown real noise map), and further replace the regularizer by $\tau\phi(\mathbf{z}) + \eta\psi(\mathbf{u})$ (regularizers for $\mathbf{z}$ and $\mathbf{u}$ weighted by $\tau$ and $\eta$ respectively, where $\tau, \eta > 0$).

3) Compared with the regularizer based on general image statistical characteristics, better results can be obtained if certain specific information of the given image is exploited to guide the denoising process. Let $\mathbf{g}$ be certain sophisticated guidance information. Then, we can improve $\phi(\mathbf{z})$ by a specific constraint $\mathcal{G}(\mathbf{z}, \mathbf{g})$ for further performance enhancement, and we call it the GC scheme.

**Proposed Model-based Denoising Method.** Before the introduction of our denoising framework, we first introduce the real noise model. Different from AWGN, real image noise is generally complex and signal-dependent. Let $\mathbf{x}_r = p(\mathbf{x}) \in \mathbb{R}^n$ be the irradiance image of RAW pixels, and $p$ be the mapping function from RGB to RAW. According to [19], the following heteroscedastic Gaussian is a reasonable approximation for real noise distribution: $\mathbf{n}(\mathbf{x}_r) = \mathbf{n}_s(\mathbf{x}_r) + \mathbf{n}_c \sim \mathcal{N}(0, u^2(\mathbf{x}_r))$. Specifically, $\mathbf{n}_c$ is a stationary noise component with variance $u_c^2$, and $\mathbf{n}_s$ is a signal-dependent noise component with space-variant variance $\mathbf{x}_r u_s^2$. Then, $\mathbf{y} = q(\mathbf{x}_r + \mathbf{n}(\mathbf{x}_r))$, where $q$ is the mapping function from RAW to RGB. Inspired by [19], the real image noise variance can be formulated by

$$u^2(\mathbf{x}_r) = \mathbf{x}_r u_s^2 + u_c^2, \tag{2}$$

where $u_s$ and $u_c$ are uniformly sampled from the ranges of [0, 0.16] and [0, 0.06]. For more details, please refer to [19]. Then, the noise map is defined as $\mathbf{u} = \sqrt{u^2(\mathbf{x}_r)} \in \mathbb{R}^n$, which is signal-dependent

and space-variant with the same size of $\mathbf{x}_r$ instead of a scalar. Next, according to the 'Analysis and Enhancement' subsection, the proposed enhanced denoising method can be given as

$$\{\hat{\mathbf{z}}, \hat{\mathbf{u}}\} = \arg\min_{\mathbf{z},\mathbf{u}} \mathcal{H}(\mathbf{z}, \mathbf{u}, \mathbf{y}) + \tau \mathcal{G}(\mathbf{z}, \mathbf{g}) + \eta\psi(\mathbf{u}), s.t., \hat{\mathbf{x}} = D(\hat{\mathbf{z}}), \tag{3}$$

where $D : \mathbb{R}^{n \cdot m} \to \mathbb{R}^{n \cdot c}$ is the LS decoding function that maps the latent embeddings $\hat{\mathbf{z}}$ to $\hat{\mathbf{x}}$.

## 2.2 Self-Correction (SC) Alternative Optimization

According to the widely used traditional alternative optimization algorithm [8, 9, 38, 29] in image restoration, Eq. (3) can be solved by splitting it into two alternative subproblems with respect to $\mathbf{z}$ (reconstruction (RE) subproblem) and $\mathbf{u}$ (degradation estimation (DE) subproblem):

$$\mathbf{z}^{(i+1)} = \arg\min_{\mathbf{z}} \mathcal{H}(\mathbf{z}, \mathbf{u}^{(i)}, \mathbf{y}) + \tau \mathcal{G}(\mathbf{z}, \mathbf{g}), \quad \mathbf{u}^{(i+1)} = \arg\min_{\mathbf{u}} \mathcal{H}(\mathbf{z}^{(i+1)}, \mathbf{u}, \mathbf{y}) + \eta\psi(\mathbf{u}), \tag{4}$$

where $i$ is the iteration number. However, the last estimates of $\mathbf{z}$ and $\mathbf{u}$ are not used in Eq. (4). For higher performance, it is better to use the last estimates for SC when estimating $\mathbf{z}$ and $\mathbf{u}$. In addition, since the estimated noise map is available during the iterations, it can be reasonable information to guide the reconstruction (*i.e.*, $\mathbf{g}$ can be obtained according to $\mathbf{u}^{(i)}$). Thus, instead of using Eq. (4), we use the SC scheme (introducing $\mathbf{u}^{(i)}$ and $\mathbf{z}^{(i)}$) and specify the form of $\mathbf{g}$, and Eq. (3) becomes:

$$\begin{cases} \mathbf{g}^{(i)} = G(\mathbf{u}^{(i)}), \\ \mathbf{z}^{(i+1)} = \arg\min_{\mathbf{z}} \mathcal{H}(\mathbf{z}, \mathbf{u}^{(i)}, \mathbf{y}) + \tau \widetilde{\mathcal{G}}(\mathbf{z}, \mathbf{g}^{(i)}, \mathbf{z}^{(i)}) = Z(\mathbf{z}^{(i)}, \mathbf{y}, \mathbf{g}^{(i)}, \mathbf{u}^{(i)}), \\ \mathbf{u}^{(i+1)} = \arg\min_{\mathbf{u}} \mathcal{H}(\mathbf{z}^{(i+1)}, \mathbf{u}, \mathbf{y}) + \eta\widetilde{\psi}(\mathbf{u}, \mathbf{u}^{(i)}) = U(\mathbf{z}^{(i+1)}, \mathbf{y}, \mathbf{u}^{(i)}), \end{cases} \tag{5}$$

where $G(\cdot)$ is the guidance information generator, $\widetilde{\mathcal{G}}(\cdot)$ becomes the joint constraint of GC and SC for $\mathbf{z}$, and $\widetilde{\psi}(\cdot)$ becomes the joint constraint of noise information and SC for $\mathbf{u}$. The two functions $Z(\cdot)$ and $U(\cdot)$ are the optimization solving processes for the $\mathbf{z}$ and $\mathbf{u}$ subproblems in Eq. (5).

Finally, $\mathbf{x}$ can be obtained by applying the LS decoding function $D$ to $\mathbf{z}$. Note that $\mathbf{u}^{(0)}$ can be directly estimated by any reasonable DE estimator $U_{\text{ini}}$ (*i.e.*, $\mathbf{u}^{(0)} = U_{\text{ini}}(\mathbf{y})$), and $\mathbf{z}^{(0)}$ can be simply set to $E(\mathbf{y})$. The overall algorithm is called SC alternative optimization based enhanced denoising (SCAOED), and is described in **Algorithm 1** of the 'Supplementary Material'.

## 3 SCAOED Driven Denoising Network

### 3.1 Implicitly Implement SCAOED Algorithm via Deep Network

It is very challenging to manually design the optimal operators in SCAOED, *e.g.* $E, D, G, U_{\text{ini}}, U, Z$. Therefore, the deep unfolding scheme is applied to implicitly implement these operators via deep network, leading to ScaoedNet. Specifically, we parameterize these operators by introducing learnable parameters $\mathbf{\Theta} = \{\mathbf{\Theta}_E, \mathbf{\Theta}_D, \mathbf{\Theta}_G, \mathbf{\Theta}_{U_{\text{ini}}}, \mathbf{\Theta}_U, \mathbf{\Theta}_Z\}$, which are learned in a discriminative manner. As shown in Fig. 2, ScaoedNet is an iterative real denoising framework in LS. Overall, **LS encoding module, LS decoding module, $G$-module, initial DE network ($\mathbf{u_{\text{ini}}}$-Net), DE network ($\mathbf{u}$-Net), and RE network ($\mathbf{z}$-Net) are used to implement $E$, $D$, $G$, $U_{\text{ini}}$, $U$, and $Z$, respectively**.

Specifically, in order to reduce the computational complexity while increasing the receptive field, the LS encoding module (the learned version of $E$) is implemented by a pixel-shuffle layer with factor $1/2$ (PS$\times 1/2$) followed by a $3 \times 3$ Conv layer, which can project the original low-dimensional image space input $\mathbf{y} \in \mathbb{R}^{c \times H \times W}$ ($H \cdot W = n$) to the high-dimensional embedding space by $E(.|\mathbf{\Theta}_E) \in \mathbb{R}^{m \times H/2 \times W/2}$ ($m$ is set to 64) as initial latent image embeddings $\mathbf{z}^{(0)}$, *i.e.* $\mathbf{z}^{(0)} = E(\mathbf{y}|\mathbf{\Theta}_E)$. In the meanwhile, $\mathbf{u}^{(0)} \in \mathbb{R}^{1 \times H \times W}$ is estimated by $\mathbf{u_{\text{ini}}}$-Net (the learned version of $U_{\text{ini}}$) to implicitly implement $U_{\text{ini}}(\mathbf{y})$. Then, GR is obtained by inputting $\mathbf{u}^{(0)}$ into the $G$-module (the learned version of $G$). Next, $\mathbf{y}, \mathbf{g}^{(0)}, \mathbf{z}^{(0)}$, and $\mathbf{u}^{(0)}$ are fed into $\mathbf{z}$-Net (the learned version of $Z$) to get $\mathbf{z}^{(1)}$. After that, $\mathbf{y}, \mathbf{u}^{(0)}$, and $\mathbf{z}^{(1)}$ are fed into the next $\mathbf{u}$-Net (the learned version of $U$) to obtain $\mathbf{u}^{(1)}$. Then, $\mathbf{g}, \mathbf{u}$ and $\mathbf{z}$ are alternatively updated by the following $G$-modules, $\mathbf{u}$-Nets, and $\mathbf{z}$-Nets, whcih implicitly implement Eq. (5). Finally, $\mathbf{z}^{(K)}$ is reconstructed by the LS decoding module (the learned version of $D$) to obtain the final image space estimate for $\mathbf{x}$, *i.e.* $\mathbf{x} = D(\mathbf{z}^{(K)}|\mathbf{\Theta}_D)$, where the LS decoding module is implemented by 'Conv-PS$\times 2$' layers. Note that the parameters of all $\mathbf{u}$-Nets, $G$-modules, and $\mathbf{z}$-Nets are respectively shared. For the reconstruction loss, we adopt the $L_1$ loss between the output and the ground-truth image. In addition, we adopt structural similarity (SSIM) loss [51] $\mathcal{L}_S$ to pay attention to image structures. For the noise map estimation loss, we adopt the $L_1$ loss between the estimated noise map and the ground-truth version. Note that the initial DE network is pre-trained

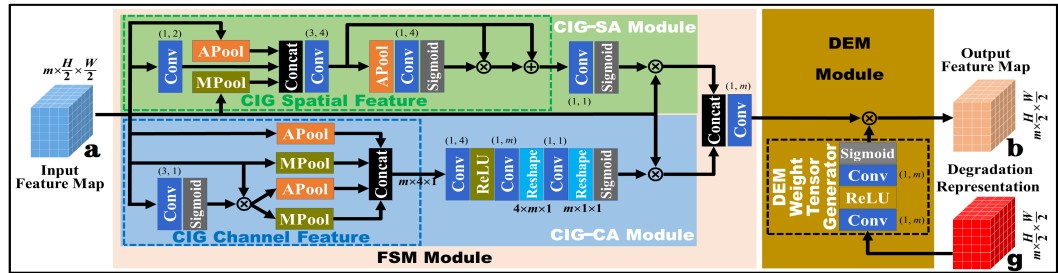

Figure 3: Architecture of the proposed FM$^2$A module consisting of FSM and DEM modules.

only with the noise map estimation loss. Let $\{\mathbf{x}_g, \mathbf{y}_g\}_{g=1}^N$ be $N$ clean-noisy training pairs, $\otimes$ be the element-wise product, and $\eta$, $\gamma$ be two positive constants. The total loss function can be given by

$$\mathcal{L}(\Theta) = \frac{1}{N}\sum_{g=1}^N(\|D(\mathbf{z}_g^{(K)}) - \mathbf{x}_g\|_1^1 + \gamma\mathcal{L}_S(D(\mathbf{z}_g^{(K)}), \mathbf{x}_g)) + \frac{\eta}{N}\sum_{i=1}^{K-1}\sum_{g=1}^N\|\boldsymbol{\kappa}_{i,g}\otimes(\hat{\mathbf{u}}_g^{(i)} - \mathbf{u}_g)\|_1^1, \quad (6)$$

where $\hat{\mathbf{u}}_g^{(i)}$ is the estimated noise map of the $i$th DE network for $\mathbf{y}_g$, and $\mathbf{u}_g$ is the ground-truth version. $\boldsymbol{\kappa}_{i,g} = \nu_i \cdot \boldsymbol{\alpha}_g$. $\nu_i$ is the weight for the $i$th DE network ($\nu_i$-s compose the geometric sequence with a common ratio $\iota$ and sum 1, i.e., $\nu_i = (\iota - 1)\iota^{i-1}/(\iota^{K-1} - 1)$). $\boldsymbol{\alpha}_g$ is the $g$th element of the indicator vector $\boldsymbol{\alpha}$ for the noise constraint. For a synthetic training pair with known noise map, $\boldsymbol{\alpha}_g = 1$. For a real training pair without known noise map, $\boldsymbol{\alpha}_g = 0$.

### 3.2 Initial Degradation Estimation (DE) Network ($\mathbf{u_{ini}}$-Net) and DE Network ($\mathbf{u}$-Net)

**Basic Architecture of $\mathbf{u_{ini}}$-Net and $\mathbf{u}$-Net.** In $\mathbf{u_{ini}}$-Net, the input $\mathbf{y}$ is first processed by a noise feature enhancement module $\mathcal{Q}$ consisting of a PS$\times 1/2$ layer, four 'Conv-ReLU' layers, and an NFAE layer, to get the enhanced noise feature map. Next, the feature map is projected to the noise map by a Conv layer and a PS$\times 2$ layer. The procedure of $\mathbf{u_{ini}}$-Net can be written as:

$$\mathbf{u}^{(0)} = U_{\text{ini}}(\mathbf{y}|\Theta_{U_{\text{ini}}}), \quad (7)$$

where $U_{\text{ini}}(.|\Theta_{U_{\text{ini}}})$ is the initial DE estimator parameterized by $\Theta_{U_{\text{ini}}}$.

The $\mathbf{u}$-Net implicitly implements DE subproblem via a deep network similar to $\mathbf{u_{ini}}$-Net. $\mathbf{u}^{(i)}$ is first downsampled by PS$\times 1/2$ followed by a Conv layer to obtain the feature map $\mathbf{u}_p^{(i)}$. In addition, $\mathbf{y}$ is input into the feature extraction module $\mathcal{Q}$ with shared parameters of $\mathcal{Q}$ in $\mathbf{u_{ini}}$-Net to implicitly enhance the non-linear fitting ability of $\mathbf{u}$-Net. Then, $\mathcal{Q}(\mathbf{y})$ is processed by a Conv layer, and further fused with $\mathbf{u}_p^{(i)}$ and $\mathbf{z}^{(i+1)}$ by a Concat layer and a $1 \times 1$ Conv layer to obtain the fused feature map. Next, four 'Conv-ReLU' layers followed by an NFAE layer are used. After that, the enhanced feature map is projected to the noise residual map by a Conv layer and a PS$\times 2$ layer. Finally, a skip connection from $\mathbf{u}^{(i)}$ to the noise residual map is added. The procedure of $\mathbf{u}$-Net can be written as:

$$\mathbf{u}^{(i+1)} = U(\mathbf{z}^{(i+1)}, \mathbf{y}, \mathbf{u}^{(i)}|\Theta_U), \quad (8)$$

where $U(.|\Theta_U)$ is the optimization solving process of the $\mathbf{u}$ subproblem parameterized by $\Theta_U$.

**Noise Feature Adaptive Enhancement (NFAE) Layer.** The feature of real image noise should have the following properties: a salient voxel with strong degradation should be much different from the voxels around it. The difference can be easily measured by the variance, and a salient voxel with a larger variance in and between channels of $\mathbf{E}_{\text{in}}^{(i+1)}$ ($\mathbf{E}_{\text{in}}^{(i+1)}$ is the input of NFAE) should have a larger weight than a non-salient voxel. Let $v_{hw}^t$ be the variance for the $hw$th voxel within the $t$th channel of $\mathbf{E}_{\text{in}}^{(i+1)}$, $Z_{hw}$ be the mean of $v_{hw}^t$-s at the location $hw$, and $Z_t$ be the mean of $v_{hw}^t$-s within the $t$th channel. We can calculate the weight without adding parameters to $\mathbf{u_{ini}}$-Net and $\mathbf{u}$-Net by

$$\mathbf{v}(t, h, w) = S(10v_{hw}^t/(Z_{hw} + \delta), 10v_{hw}^t/(Z_t + \delta)), \quad (9)$$

where $S(\cdot, \cdot)$ is the product of the Sigmoid results of its two inputs, i.e., $S(a, b) = \text{Sigmoid}(a) \cdot \text{Sigmoid}(b) = 1/[(1 + e^{-a}) \cdot (1 + e^{-b})]$, where $e$ is the Euler's number. $\delta$ ($10^{-16}$) is a small positive factor for numeral stability. Thus, the output of NFAE can be obtained by $\mathbf{v} \otimes \mathbf{E}_{\text{in}}^{(i+1)}$.

### 3.3 Guidance Representation Module (G-Module) and Reconstruction Network ($\mathbf{z}$-Net)

Since the estimated noise map is available during the iterations, it can be reasonable information to guide the reconstruction. Therefore, $\mathbf{g}^{(i)}$ in Eq. (5) is simply implemented by the encoded degradation

representation obtained by the **G**-Module in Fig. 2. The procedure of $G$-Module can be written as:

$$\mathbf{g}^{(i)} = G(\mathbf{u}^{(i)}|\boldsymbol{\Theta}_G), \tag{10}$$

where $G(.|\boldsymbol{\Theta}_G)$ is the guidance information generator parameterized by $\boldsymbol{\Theta}_G$.

**z**-Net is designed to restore the latent **z**, which implicitly implement the **z** subproblem via a deep network. As shown in Fig. 2, with the degradation representation $\mathbf{g}^{(i)}$, we further propose the FM$^2$A module shown in Fig. 3 to adjust the feature map according to the internal and external information. In detail, we adopt the FSM module to capture spatial and channel information of image features themselves, and the DEM module to dynamically recalibrate the image features according to the guidance information. By applying FM$^2$A to a residual unit, FM$^2$ARB is proposed to comprehensively modulate the residual features. After that, an skip connection from the Conv layer after $\mathbf{z}^{(i)}$ to the recalibrated feature by DEM is added. Finally, the feature map is adjusted by a Conv layer, and then added with $E(\mathbf{y})$. The procedure of **u**-Net can be written as:

$$\mathbf{z}^{(i+1)} = Z(\mathbf{z}^{(i)}, \mathbf{y}, \mathbf{g}^{(i)}, \mathbf{u}^{(i)}|\boldsymbol{\Theta}_Z), \tag{11}$$

where $Z(.|\Theta_Z)$ is the optimization solving process of the **z** subproblem parameterized by $\boldsymbol{\Theta}_Z$.

### 3.4 Feature Multiple-Modulation Attention

As shown in Fig. 3, a novel FM$^2$A module consisting of FSM and DEM modules is proposed.

**Feature Self-Modulation (FSM).** Attention mechanism is very important in image restoration tasks. Although self-attention mechanism [42] has been widely used recently, because of its high parameter number and complexity, the spatial/channel attention mechanism is finally adopted in our work for a balance between performance and complexity. Different from the traditional spatial attention (SA) and channel attention (CA) mechanisms [50], we introduce complementary feature information into the proposed FSM module to enhance the recalibration ability in and between channels of the feature map. It consists of complementary information guided (CIG) SA (*i.e.*, CIG-SA) and CIG-CA. For CIG-SA, CIG spatial feature is first achieved by channel-wise modulation, and then used to calculate the spatial weights. For CIG-CA, CIG channel feature is first achieved by space-wise modulation, and then used to calculate the channel weights. After both the spatial and channel weights are obtained, they are respectively used to recalibrate the input feature map, and then their results are further fused. Note that by removing the CIG scheme in FSM, it becomes a spatial-channel attention (SCA) based on traditional SA and CA. **For more details about CIG-SA and CIG-CA, please refer to the 'Supplementary Material'.**

**Degradation External-Modulation (DEM).** Because of the complexity of real degradation, it is a great challenge to estimate the clean image directly by the usual deep network. As analyzed in section 2, it is promising to exploit the guidance information (degradation representation) **g** to guide the reconstruction. Therefore, to exploit **g** in the implementation of GC, the feature map processed by the FSM module is further recalibrated by the tensor generated by the DEM weight tensor generator.

For the input feature map **a**, by applying both FSM and DEM, the FM$^2$A module can synchronously realize external GC and the internal important information mining to get the output feature map **b**:

$$\mathbf{b} = C_{\text{fsm}}(\mathbf{a}) \otimes C_{\text{dem}}^{\text{att}}(\mathbf{g}), \tag{12}$$

where $C_{\text{dem}}^{\text{att}}$ is the DEM weight tensor generator, and $C_{\text{fsm}}$ is the FSM operator.

## 4 Experiments

### 4.1 Training and Testing

To train ScaoedNet, Div2K [41] is adopted to generate the synthetic clean-noisy image pairs according to the noise model in subsection 2.1, and the real clean-noisy image pairs from SIDD Medium Dataset [2] and RENOIR Dataset [4] are also used for training. Random rotations of $90°$, $180°$, $270°$, and horizontal flipping are performed for each training sample to achieve data augmentation. We empirically set $K = 5$, $\eta = 0.5$, $\gamma = 0.25$, $\iota = 4$, and $T = 16$. Network details (*e.g.*, channel number, kernel size) are provided in Figs.2-3, and the default number $m$ is set to 64. Adam [23] is used as the optimizer for the network model with default settings. During the training, we initially set the patch size to $64 \times 64$, the mini-batch size to 64, and the learning rate to $10^{-4}$. For fine-tuning, we set the patch size to $256 \times 256$, the mini-batch size to 4, and the learning rate to $10^{-5}$. We use PyTorch to implement ScaoedNet and train it with an Nvidia GeForce RTX 3090 GPU. Furthermore,

Table 1: The average PSNR(dB)/SSIM results on DnD benchmark dataset.

| Method | Blind/Non-Blind | PSNR↑ | SSIM↑ |
|---|---|---|---|
| CBM3D[14] | Non-Blind | 34.51 | 0.8507 |
| TNRD[11] | Non-Blind | 33.65 | 0.8306 |
| DnCNN[52] | Blind | 32.43 | 0.7900 |
| FFDNet[53] | Non-Blind | 37.61 | 0.9415 |
| DCDicL[57] | Non-Blind | 35.90 | 0.9150 |
| CBDNet[19] | Blind | 38.06 | 0.9421 |
| VDN[48] | Blind | 39.38 | 0.9518 |
| RIDNet[5] | Blind | 39.25 | 0.9528 |
| AINDNet[22] | Blind | 39.37 | 0.9505 |
| InvDN[27] | Blind | 39.57 | 0.9522 |
| DANet[49] | Blind | 39.47 | 0.9548 |
| DeamNet[36] | Blind | 39.63 | 0.9531 |
| ScaoedNet | Blind | **40.12** | **0.9603** |
| ScaoedNet† | Blind | **40.17** | **0.9597** |

Table 2: The average PSNR(dB)/SSIM results on SIDD benchmark and validation datasets.

| Method | Blind/ Non-Blind | SIDD benchmark | | SIDD validation | |
|---|---|---|---|---|---|
| | | PSNR↑ | SSIM↑ | PSNR↑ | SSIM↑ |
| CBM3D[14] | Non-Blind | 25.65 | 0.685 | 31.75 | 0.7061 |
| TNRD[11] | Non-Blind | 24.73 | 0.643 | 26.99 | 0.7440 |
| DnCNN[52] | Blind | 23.66 | 0.583 | 26.20 | 0.4414 |
| FFDNet[53] | Non-Blind | 29.30 | 0.694 | 26.21 | 0.6052 |
| DCDicL[57] | Non-Blind | 33.68 | 0.860 | 33.76 | 0.8171 |
| CBDNet[19] | Blind | 33.28 | 0.868 | 30.83 | 0.7541 |
| VDN[48] | Blind | 39.26 | 0.955 | 39.29 | 0.9109 |
| RIDNet[5] | Blind | 37.87 | 0.943 | 38.76 | 0.9132 |
| AINDNet[22] | Blind | 38.95 | 0.952 | 38.96 | 0.9123 |
| InvDN[27] | Blind | 39.28 | 0.955 | 38.30 | 0.9064 |
| DANet[49] | Blind | 39.25 | 0.955 | 39.30 | 0.9164 |
| DeamNet[36] | Blind | 39.35 | 0.955 | 39.40 | 0.9169 |
| ScaoedNet | Blind | **39.44** | **0.956** | **39.48** | **0.9186** |
| ScaoedNet† | Blind | **39.48** | **0.957** | **39.52** | **0.9187** |

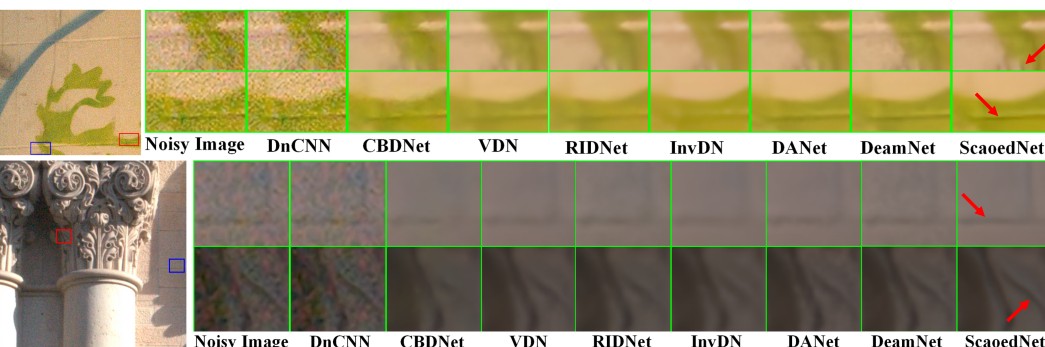

Figure 4: Real denoising results on DnD Benchmark dataset.

PSNR and SSIM metrics are utilized for performance evaluation on the following real noisy datasets: DnD benchmark [35], SIDD benchmark [2], and SIDD validation dataset [2]. The near noise-free images of DnD and SIDD benchmarks are not publicly available. However, *we can submit the denoised image to the DnD and SIDD online servers to obtain the PSNR/SSIM results.* **Due to the limited space, please refer to the 'Supplementary Material' for more experimental results.**

### 4.2 Real-World Denoising on DnD

DnD consists of 50 pairs of real clean-noisy scenes, where 20 smaller images of size $512 \times 512$ are extracted from each scene. However, the corresponding ground-truth images are not available for users. In this subsection, we evaluate the performance of ScaoedNet on the DnD benchmark. Specifically, 12 competing methods are tested, including CBM3D [14], TNRD [11], DnCNN [52], FFDNet [53], DCDicL [57], CBDNet [19], VDN [48], RIDNet [5], AINDNet [22], InvDN [27], DANet [49], and DeamNet [36]. Note that AINDNet(TF) is tested in the experiment since it can get the best overall performance on both DnD and SIDD among all AINDNet models according to [22, 36]. DCDicL [57] gets lower results because it is designed for AWGN. To better show the denoising performance of ScaoedNet, the results of the self-ensemble [40] version denoted by the super script † are also reported. Table 1 and Fig. 4 show the objective and visual results. **Due to the limited space, please enlarge the figures on the screen for better comparison.** According to the results, ScaoedNet significantly outperforms these competing methods in PSNR/SSIM results. In addition, these methods prone to generate either over-smooth results or distorted results with remaining noise. On the contrary, ScaoedNet can provide a clean output image with robust noise removal, effective artifacts suppression, and excellent image edges preservation.

### 4.3 Real-World Denoising on SIDD

We further evaluate the generalization and performance of ScaoedNet on SIDD captured by five representative smartphones. Only 320 clean-noisy image pairs (SIDD Medium Dataset) are provided for training. For testing, SIDD benchmark only with the noisy images is provided. In addition, SIDD validation consists of 1280 clean-noisy image pairs can also be used. In this subsection, extensive experiments are carried out on SIDD validation and SIDD benchmark to evaluate the superiority of

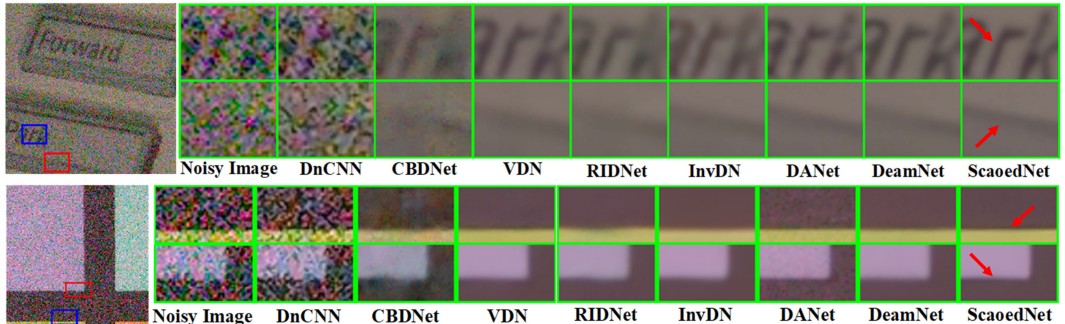

Figure 5: Real denoising results on SIDD validation dataset.

Table 3: Results on SIDD validation for $K$-s.

| $K$ | 1 | 2 | 3 | 4 | 5 | 6 |
|------|------|------|------|------|------|------|
| PSNR | 39.24 | 39.32 | 39.40 | 39.45 | 39.48 | 39.48 |
| SSIM | 0.9170 | 0.9177 | 0.9182 | 0.9184 | 0.9186 | 0.9187 |

Table 4: Results on SIDD validation for $T$-s.

| $T$ | 4 | 8 | 12 | 16 | 20 |
|------|------|------|------|------|------|
| PSNR | 39.10 | 39.26 | 39.40 | 39.48 | 39.49 |
| SSIM | 0.9158 | 0.9169 | 0.9179 | 0.9186 | 0.9187 |

ScaoedNet. Table 2 and Fig. 5 illustrate the quantitative and visual comparisons between the previous methods and ScaoedNet. We can notice that ScaoedNet achieves the best results compared to other competing methods in terms of both the objective evaluation and the visual performance.

## 4.4 Influence of Parameters $K$ and $T$

Since the iteration stage number $K$ and the FM$^2$ARB number $T$ are crucial for denoising performance, the PSNR/SSIM evolutions versus $K$ and $T$ are analyzed. Specifically, we train six ScaoedNet variants by setting $K = 1, 2, 3, 4, 5, 6$ and five ScaoedNet variants by setting $T = 4, 8, 12, 16, 20$. The PSNR/SSIM results on SIDD validation dataset are provided in Tables 3 and 4. According to the results, when $K = 1$, ScaoedNet can already achieve promising denoising performance with 39.24dB/0.9170. When $K$ reaches 5, PSNR/SSIM further converge to 39.48dB/0.9186, which indicates that $G$-Module, $\mathbf{u}$-Net, and $\mathbf{z}$-Net have learned to cooperate with each other in this case. For $T$, we can see that $T = 16$ is significantly better than $T = 4, 8, 12$, and there is little PSNR/SSIM improvements when $T$ is larger than 16. These suggest that $K = 5$ and $T = 16$ are good choices.

## 4.5 Ablation Study

**Ablation on LS.** By removing the LS encoding and decoding modules from ScaoedNet, we can obtain the image space denoising network. Because the Shuffle operator with factor 1/2 is used, the output of each RE network in the image space version have only 12 channels for color images and 4 channels for gray images. We can see from Fig. 6 that the PSNR/SSIM results of the version without LS on SIDD validation dataset are only 39.19dB/0.9163, and the performance reaches to 39.48dB/0.9186 by adding LS. Therefore, the LS scheme is essential for the denoising performance.

These reasons make the LS scheme essential for the performance. In Sec. 4.5 'Ablation Study', 'w/o LS' corresponds to the network implementation without LS, and the performance decreases significantly in this case, which verifies our analyses. In addition, by visualizing the features after $E$, we find it can well separate the image signal related features and noise related components, which also makes it easier to denoise complex real noise.

**Ablation on SC.** In our method, the SC scheme is proposed for enhancing the optimization results. The ablation study on SC is conducted and the performance comparisons are shown in Fig. 6. Note that the version without SC means $\mathbf{u}^{(i)}$ and $\mathbf{z}^{(i)}$ are respectively removed when estimating $\mathbf{u}^{(i+1)}$ and $\mathbf{z}^{(i+1)}$. We can see that the PSNR/SSIM results decrease from 39.48dB/0.9186 to 39.22dB/0.9170 without SC. This observation indicates that the SC scheme is an effective scheme in ScaoedNet.

**Ablation on GC.** For further performance enhancement, the GC scheme is proposed. Note that the version without GC means the branches related to $\mathbf{g}^{(i)}$-s are removed. In other words, $G$-Module and DEMs are removed. We can conclude from Fig. 6 that the PSNR/SSIM results decrease 0.13dB/0.0015 without GC. This observation indicates the superiority of GC in ScaoedNet.

**Ablation on Attention.** The proposed attention module can synchronously realize external-internal important information mining. Fig. 6 shows that, by removing FM$^2$A, the PSNR/SSIM will decrease

| Method | $L_1$ distance↓ | PSNR↑ |
|---|---|---|
| with NFAE | 5.80 | 33.16 |
| without NFAE | 5.97 | 32.53 |
| CBDNet[19] | 6.11 | 32.30 |
| AINDNet [22] | 6.11 | 32.92 |
| PRIDNet [56] | 6.19 | 33.01 |

Table 5: The effectiveness of the NFAE layer.

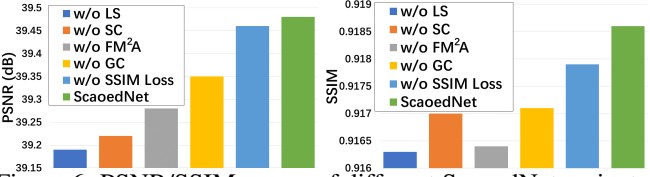

Figure 6: PSNR/SSIM scores of different ScaoedNet variants.

from 39.48dB/0.9186 to 39.28dB/0.9164. Consequently, the effectiveness of FM$^2$A is verified. More details of FSM and DEM are provided in 'Supplementary Material'.

**Ablation on SSIM Loss.** To verify the influence of the SSIM loss in our method, we remove it from the loss function in Eq. 6. We can see from Fig. 6 that PSNR/SSIM of the one without SSIM loss will decrease slightly to 39.46dB/0.9179. We should note that, even without the SSIM loss, our method can also achieve higher performance than other comparisons baselines.

### 4.6 Study on NFAE layer

The proposed parameter-free NFAE layer is an important component in ScaoedNet. To show the effectiveness of NFAE, we use BSD68 [37] and Kodak24 [1] to synthesize the noisy images according to the noise model in subsection 2.1. Then, we build 'noisy image and noise map' pairs as the testing dataset. $\mathbf{u}_{\text{ini}}$-Net with NFAE, $\mathbf{u}_{\text{ini}}$-Net without NFAE, and the noise estimation networks from CBDNet [19], AINDNet [22], and PRIDNet [56] are used for comparisons. The PSNR result and the $L_1$ distance $\sum_{g=1}^{V} \|\hat{\mathbf{u}}_g - \mathbf{u}_g\|_1^1/(H_g W_g V)$ ($V$ is the number of the test images, and $H_g, W_g$ are the image height and width of the $g$th test image) between the estimated $\hat{\mathbf{u}}_g$ and the ground-truth noise map $\mathbf{u}_g$ are reported in Table 5. Note that during the calculations of the $L_1$ distance and PSNR, $\mathbf{u}$ is scaled to lie in the range of [0 255]. The results show that the NFAE layer is essential for the estimation performance, and the network with NFAE can obtain more accurate estimates than the noise estimation networks from CBDNet [19], AINDNet [22], and PRIDNet [56].

### 4.7 Parameter Number

In this subsection, we report the network parameter numbers *vs.* the average PSNRs/SSIMs of the competing methods in Fig. 7. We can see that ScaoedNet has a moderate parameter number, which is significantly lower than CBDNet [14], VDN [48], AINDNet [22], and DANet [49]. Furthermore, although other methods have smaller parameter numbers than ScaoedNet, their objective performances are much lower. In summary, our ScaoedNet is effective with a moderate network parameter number.

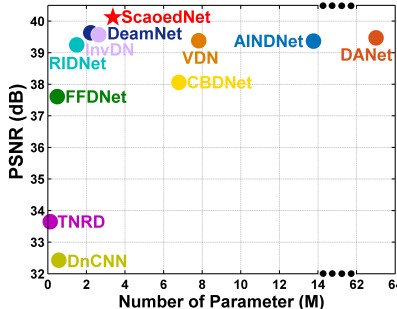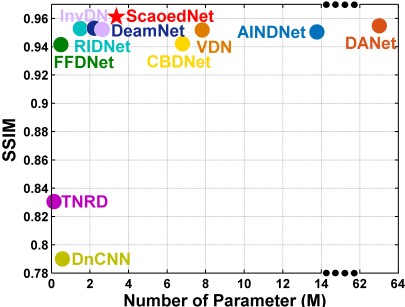

Figure 7: The numbers of parameter *vs.* average PSNR/SSIM values of different models on DnD.

## 5 Conclusion

In this paper, we propose a new denoising network for real noise images. Although many denoising networks have been proposed, most of them are limited to AWGN and ignore the valuable achievements of the classic denoising methods. In contrast, we first propose a novel model-based denoising cost function, and then solve it by the proposed SC alternative optimization algorithm. After that, a deep unfolding denoising network is proposed according to the optimization process. Our denoising network combines some valuable achievements of the classical denoising methods. Extensive experiments verify that our method achieves excellent performance on real image denoising.

## Broader Impact

As an important computer vision task, image denoising has largely benefited the society in various areas and has no negative impact yet. The proposed method could further improve the performance especially in the challenging real-world cases where the noise information is unknown. This work has no negative impact on the ethical and societal aspects.

## Acknowledgements

This work was supported by the National Natural Science Foundation of China under Grant 62171304, and in part by the Key Research and Development Program of Sichuan Province under Grant 2022YFS0098.

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
