# OpenReview forum: "Enhanced Latent Space Blind Model for Real Image Denoising via Alternative Optimization"
_NeurIPS.cc/2022/Conference — NeurIPS 2022 Accept_

### Official Review · Reviewer_XEky · 2022-07-09

**Rating:** 6
**Confidence:** 4
**Soundness:** 3 good
**Presentation:** 4 excellent
**Contribution:** 3 good

**Summary:**

This paper proposes an image denoising network based on latent space self-correction alternative optimization. A deep unfolding network implements alternative optimization, and state of the art denoising performance is achieved on multiple benchmark datasets.

**Questions:**

1. Would raise score if inference complexity/time is on par with other state of the art methods.
2. Is the proposed model based framework applicable to other image restoration problems, e.g. deblurring, deraining, etc. ? This paper only discusses denoising and the value of the proposed methodology is therefore limited.

**Strengths And Weaknesses:**

Strengths:
- the proposed method is backed with classic model based image restoration theory that provides strong regularization on the network design, which allows the proposed method to achieve state of the art performance with a modest model size.
- implementation detail of the proposed network is well documented for reproduction
- extensive ablation study is conducted to analysis each individual component of the proposed network architecture

Weakness:
- The experimental evaluation section lacks complexity and time cost analysis and comparison
- section 3 is filled with an overwhelming amount of implementation detail that are better suited for supplementals. It would be better to put emphasis on theoretic/algorithmic reasoning & insights, and put implementation details in section 4 or a dedicated subsection in section 3. Right now section 3 provides very limited insight into *how* the network implements and enforces those ideas.

---

> ### Author Response · Authors · 2022-08-02
> **Responses to Reviewer XEky**
>
> **Comment:**
> We thank the reviewer for the valuable comments and approval for our work.
>
> >**The experimental evaluation section lacks complexity and time cost analysis and comparison.**
>
> The complexity and time for $512\times512$ images have been reported in Sec. ‘S8. Computational Complexity and Inference Time’ of the ‘Supplementary Material’. Moreover, for $256\times256$ images, the FLOPs and inference times of ScaoedNet with 1 stage, 3 stages and 5 stages are 53G/0.02s, 160G/0.05s, and 268G/0.08s, respectively. For the second-best method DeamNet, it is 146G/0.05s. We can further reduce the complexity by using the multi-scale encoder-decoder based UNet architecture similar to DeamNet with four scales, where each encoder or decoder can consist of two FM$^2$ARBs. Then, the FLOPs in this case for 256$\times$256 images will reduce to about 130G. We will add more analyses in the revised version.
>
> >**Section 3 is filled with an overwhelming amount of implementation detail that are better suited for supplementals. It would be better to put emphasis on theoretic/algorithmic reasoning & insights, and put implementation details in section 4 or a dedicated subsection in section 3.**
>
> Thank you for your suggestion. We will pay more attention to the clarity of our paper in the revised version. Specifically, we will put the implementation details into Sec. 4 and the ‘Supplementary Material’, and add more theoretic/algorithmic reasoning and insights in Sec. 3, making it easier for readers to follow.
>
> >**Is the proposed model based framework applicable to other image restoration problems, e.g. deblurring, deraining, etc. ? This paper only discusses denoising and the value of the proposed methodology is therefore limited.**
>
> As discussed in S11 ‘Limitations and Future Works’ of the ‘Supplementary Material’, the method can be potentially applied to other image restoration (IR) problems. We take image deblurring problem as an example. Image deblurring can be expressed as:
>
> $\mathbf{y}=\mathbf{x}*\mathbf{k}+\mathbf{n}$
>
> where $\mathbf{y}$ is the degraded image, $\mathbf{x}$ is the ground-truth image, $\mathbf{k}$ is the blur kernel, $\mathbf{n}$ is the noise, and $*$ denotes the convolutional operation. Then, Eq. (3) in the manuscript becomes:
>
> $\begin{Bmatrix} \hat{\mathbf{z}},\hat{\mathbf{u}},\hat{\mathbf{k}} \end{Bmatrix}=\arg\min\_{\mathbf{z},\mathbf{u},\mathbf{k}} \mathcal{H}(\mathbf{z},\mathbf{u},\mathbf{k},\mathbf{y})+\tau\mathcal{G}(\mathbf{z},\mathbf{g})+\eta\_1\psi(\mathbf{u})+\eta\_2\phi(\mathbf{k}), s.t., \hat{\mathbf{x}}=D(\hat{\mathbf{z}})$
>
> where $\tau, \eta\_1, \eta\_2$ are the weights for the regularizers $\mathcal{G}(\mathbf{z},\mathbf{g})$ (guidance constraint, GC), $\psi(\mathbf{u})$ (noise map prior), and $\phi(\mathbf{k})$ (blur kernel prior). By using our self-correction (SC) alternative optimization, we can obtain
>
> $\begin{cases}
> \mathbf{g}^{(i)} = G(\mathbf{u}^{(i)}, \mathbf{k}^{(i)}),\\\ \mathbf{z}^{(i + 1)} = \arg\min_{\mathbf{z}} \mathcal{H}(\mathbf{z}, \mathbf{u}^{(i)},\mathbf{k}^{(i)}, \mathbf{y})+\tau\widetilde{\mathcal{G}}(\mathbf{z},\mathbf{g}^{(i)}, \mathbf{z}^{(i)}),\\\ \mathbf{u}^{(i+1)}=\arg\min_{\mathbf{u}}\mathcal{H}(\mathbf{z}^{(i+1)}, \mathbf{u},\mathbf{k}^{(i)},\mathbf{y})+\eta\_1\widetilde{\psi}(\mathbf{u}, \mathbf{u}^{(i)}),\\\ \mathbf{k}^{(i+1)}=\arg\min_{\mathbf{k}}\mathcal{H}(\mathbf{z}^{(i+1)}, \mathbf{u}^{(i+1)},\mathbf{k},\mathbf{y})+\eta\_2\widetilde{\phi}(\mathbf{k}, \mathbf{k}^{(i)})
> \end{cases}$
>
> where $G(\cdot)$ is the guidance information generator, $\widetilde{\mathcal{G}}(\cdot)$ becomes the joint constraint of GC and SC for $\mathbf{z}$, $\widetilde{\psi}(\cdot)$ becomes the joint constraint of noise information and SC for $\mathbf{u}$, and $\widetilde{\phi}(\cdot)$ becomes the joint constraint of blur kernel and SC for $\mathbf{k}$.
>
> By comparing this equation with Eq. (5) in the manuscript, we can find the difference between the denoising and deblurring using our method is: in addition to the estimation of $\mathbf{z}$, $\mathbf{u}$, we have to estimate the blur kernel information $\mathbf{k}$ and construct the new guidance information $\mathbf{g}$ for deblurring. How to construct the $\mathbf{k}$ estimation module and the guidance information generator for deblurring is very important. Other image restoration problems can be solved similarly. This is also one of our future works. We will add corresponding illustrations and analyses in the revised version.
>
> ```
> We hope that the responses alleviate the reviewer's concerns. We are happy to answer any additional questions the reviewer has.
> ```

---

### Official Review · Reviewer_JAMB · 2022-07-10

**Rating:** 5
**Confidence:** 4
**Soundness:** 3 good
**Presentation:** 3 good
**Contribution:** 3 good

**Summary:**

In this paper, the authors proposed a method for real image denoising. The authors designed the enhanced model-based denoising cost function by introducing latent space, noise information and guidance constraint. And then implement the proposed self-correction alternative optimization algorithm to minimize cost function via deep network. Besides, for higher performance, some optimized modules of subnetwork are proposed, including noise feature adaptive enhancement (NFAE) layer and feature multi-modulation attention residual block (FM2ARB).

**Questions:**

1. In ‘Analysis and Enhancement’ part, the author represents that LS encoding function can obtain high-dimensional image embeddings instead of limited squared error in low-dimensional image space. However, no clear theoretical reason and significance are indicated.
2. As mentioned in the article, LS scheme is the essential for the performance of this work. However, there is still no enough explicit explanation for its ablation study.
3. The author has already chosen the SSIM as one evaluation, why repeatedly adopt SSIM loss in the total loss function?
4. Can author provide more comparison with other noise estimation network to further verify the effectiveness of NFAE layer?
5. Authors also can compare the proposed method with some more advanced traditional methods, e.g.: Exemplar-Based Denoising: A Unified Low-Rank Recovery Framework, IEEE Transactions on Circuits and Systems for Video Technology, 2020; Deep Convolutional Dictionary Learning for Image Denoising，CVPR, 2021.

**Limitations:**

In this paper, the method that authors proposed makes full use of the advantages of classical denoising methods and deep network, which no longer sticks to AWGN and contributes to real image denoising to a certain extent.

**Strengths And Weaknesses:**

Motivated by the advances in deep networks and relying on the rich body of the model based methods, authors propose a novel enhanced latent space (LS) blind model based deep unfolding network for real image denoising. Extensive experiments verify that this method achieves excellent performance on real image denoising.
However, authors should pay more attention to clarity of the paper. For example, the abstract of this paper is too mess to emphasis the most important novelty. Author needs to write it in brief and appropriately.
Besides, the layout and the visualization of some figures also need to be attached.
e.g., the Figure 1 is not mentioned in the text but shown. In Fig.2, the name of subnetworks cannot be corresponded to the figure note. Readers still need to find each subnetwork’s name through article.

---

> ### Author Response · Authors · 2022-08-02
> **Responses to Reviewer JAMB**
>
> **Comment:**
> We thank the reviewer for the constructive comments.
>
> >**Authors should pay more attention to clarity, e.g., the abstract and some figures.**
>
> We will improve the clarity. For example, we will make the abstract more appropriate, and carefully revise the layout and the visualization of figures. We will add “As shown in Fig. 1, the whole architecture of ScaoedNet achieves the best promising visual performance when compared to other denoising methods.” in Sec. 1. For Fig. 2, we will add the network names (such as $\mathbf{u}_{\text{ini}}$-Net, $\mathbf{u}$-Net, and $\mathbf{z}$-Net) into its sub-title.
>
> >**Clear theoretical reason and significance for LS are not indicated. In addition, the LS scheme is the essential for the performance, but there is no enough explicit explanation for its ablation study.**
>
> Theoretically, without using LS, Eq. (5) becomes:
>
> \begin{cases}
> \mathbf{g}^{(i)}=G(\mathbf{u}^{(i)}),\\\ \mathbf{x}^{(i+1)}=\arg\min_{\mathbf{x}}\mathcal{H}(\mathbf{x}, \mathbf{u}^{(i)}, \mathbf{y})+\tau\widetilde{\mathcal{G}}(\mathbf{x},\mathbf{g}^{(i)}, \mathbf{x}^{(i)}),\\\ \mathbf{u}^{(i+1)}=\arg\min_{\mathbf{u}}\mathcal{H}(\mathbf{x}^{(i+1)}, \mathbf{u},\mathbf{y})+\eta\widetilde{\psi}(\mathbf{u}, \mathbf{u}^{(i)}).
> \end{cases}
>
> The denoising is performed in the pixel domain to directly reconstruct $\mathbf{x}\in\mathbb{R}^{n\cdot c}$. It will lead to a $\mathbf{z}$-Net with a very low input/output channel dimension $c$, and thus the information flow in the unfolding network will be reduced.
>
> As shown in the original Eq. (5), by using LS, denoising is performed in LS to reconstruct $\mathbf{z}\in{ \mathbb{R}^{n\cdot m}}$, where $m$ is much larger than $c$. Thus, the $\mathbf{z}$-Net has more sufficient representation ability than the one without LS, and can achieve higher performance. In addition, the channel number for information flow between $\mathbf{z}$-Nets is larger compared to the version without LS, and thus the network information flow is enhanced.
>
> These reasons make the LS scheme essential for the performance. In Sec. 4.5 ‘Ablation Study’, ‘w/o LS’ corresponds to the network implementation without LS, and the performance decreases significantly in this case, which verifies our analyses. In addition, by visualizing the features after $E$, we find it can well separate the image signal related features and noise related components, which also makes it easier to denoise complex real noise.
>
> These analyses will be added in the revised version.
>
> >**The author has already chosen the SSIM as one evaluation, why repeatedly adopt SSIM loss in the total loss function?**
>
> Combining the $L_1$ or $L_2$ loss with the SSIM loss is very common in image restoration. E.g., with SSIM evaluation, [*1, *2] also use the SSIM loss. Specifically, the reconstruction loss in Eq. (6) contains two parts: 1) the $L_1$ loss between the output and the ground-truth image, to ensure the pixel-level similarity; 2) the SSIM loss $\mathcal{L}_{\text{S}}$, to pay attention to image structure preservation. We have analyzed the role of the SSIM loss in Sec. 4.5. The results show that it can slightly improve performance. But even without it, our method can still achieve good performance.
>
> [*1] Memory-efficient Hierarchical Neural Architecture Search for Image Denoising, CVPR 2020
>
> [*2] Invertible Denoising Network: A Light Solution for Real Noise Removal, CVPR 2021
>
> >**Can author provide more comparison with other noise estimation network to further verify the effectiveness of NFAE layer?**
>
> We add the comparison with other noise estimation networks from AINDNet [*3] and PRIDNet [*4]:
>
> |Method|$L_1$ Distance$\downarrow$|PSNR$\uparrow$
> :-:|:-:|:-:
> with NFAE|5.80|33.16
> AINDNet|6.11|32.92
> PRIDNet|6.19|33.01
>
> The results show that the performance of our method with NFAE is still the best. We will add these results in the revised version.
>
> [*3] Transfer Learning from Synthetic to Real-Noise Denoising with Adaptive Instance Normalization, CVPR 2020
>
> [*4] Pyramid Real Image Denoising Network, VCIP 2019
>
> >**Authors can compare the proposed method with more advanced traditional methods, e.g., PNMM; DCDicL.**
>
> Since the code of PNMM is not provided, we will cite it in the revised version. For DCDicL, we have tested it for real noisy images by the provided model. Since the additive white Gaussian noise (AWGN) level is needed as an input, we empirically set it to 75 for the best performance:
>
> |Method|SIDD Validation|SIDD Benchmark|DnD Benchmark
> :-:|:-:|:-:|:-:
> DCDicL|33.76/0.8171|33.68/0.860|35.90/0.9150
> Ours|39.48/0.9186|39.44/0.956|40.12/0.9603
>
> DCDicL gets lower results because it is designed for AWGN. By combining our real noise map estimation network with DCDicL, and then retraining it for real noise, better results may be obtained. We will analyze these in the revised version.
>
> ```
> We hope that the responses alleviate the reviewer's concerns. We are happy to answer any additional questions the reviewer has.
> ```

---

### Official Review · Reviewer_92hr · 2022-07-11

**Rating:** 5
**Confidence:** 4
**Soundness:** 3 good
**Presentation:** 2 fair
**Contribution:** 2 fair

**Summary:**

This paper proposed an enhanced latent space blind model based deep unfolding network, namely ScaoedNet, for complex real image denoising.  Experiments demonstrate the superiority of ScaoedNet over many SOTA methods.

**Questions:**

Please clarify the difference between you and [1] on how to use the distinguishable space (named latent space or subspace), and the superiority of your usage.

**Limitations:**

See in weakness.

**Strengths And Weaknesses:**

Strength:

1. An enhanced model-based denoising cost function is proposed to implicitly optimize the ScaoedNet, which addresses the challenging task of manually designing the optimal operators and making the optimal process more interpretable.

2. The effective NFAE layer leads to better results without increasing extra parameters.

Weaknesses:

1. The description of the total loss is not clear enough.

2. Map noise image to space that better distinguish noise has been proposed in [1] and therefore cannot be made as a prominent contribution.

[1] 1. Cheng, Y. Wang, H. Huang, D. Liu, H. Fan, and S. Liu. Nbnet: Noise basis learning for 356 images denoising with subspace projection. In IEEE Conference on Computer Vision and Pattern Recognition (CVPR), pages 4896–4906, Jun. 2021.

---

> ### Author Response · Authors · 2022-08-02
> **Responses to Reviewer 92hr**
>
> **Comment:**
> We thank the reviewer for the thoughtful and detailed comments.
>
> >**The description of the total loss is not clear enough.**
>
> According to Eq. (6), our total loss is as follows:
>
> $\mathcal{L}(\mathbf{\Theta})=\frac{1}{N}\sum_{g=1}^{N}(\Vert D(\mathbf{z}\_g^{(K)})-\mathbf{x}\_g\Vert_{1}^1+\gamma\mathcal{L}\_{\text{S}}(D(\mathbf{z}\_g^{(K)}), \mathbf{x}\_g))+\frac{\eta}{N}\sum_{i=1}^{K-1}\sum_{g=1}^{N}\Vert \kappa\_{i,g}\otimes(\hat{\sigma}\_{g}^{(i)}-\sigma(\mathbf{x}\_g) )\Vert_{1}^1$
>
> It contains two main parts: the reconstruction loss and the noise map estimation loss.
>
> The reconstruction loss， i.e., $\mathcal{L}(\mathbf{\Theta})=\frac{1}{N}\sum_{g=1}^{N}(\Vert D(\mathbf{z}\_g^{(K)})-\mathbf{x}\_g\Vert_{1}^1+\gamma\mathcal{L}\_{\text{S}}(D(\mathbf{z}\_g^{(K)}), \mathbf{x}\_g))$，further contains two sub-parts: 1) $L_1$ loss between the output and the ground-truth image to ensure their similarity in pixel-level; 2) the structural similarity loss $\mathcal{L}_{\text{S}}$ to pay attention to image structures. In other words, in the reconstruction loss, both the pixel-level constraint and the structure-level constraint are considered simultaneously to ensure the quality of reconstruction.
>
> The noise map estimation loss is $\frac{\eta}{N}\sum_{i=1}^{K-1}\sum_{g=1}^{N}\Vert \kappa\_{i,g}\otimes(\hat{\sigma}\_{g}^{(i)}-\sigma(\mathbf{x}\_g) )\Vert_{1}^1$. That means the estimated noise maps of the first to $(k-1)$th degradation estimation (DE) networks are constrained to be close to the ground-truth one.
>
> The weight of each stage is determined by $\kappa\_{i,g}=\nu_i\cdot\alpha\_{g}$, where $\nu_i$ is the weight for the $i$th DE network. Considering that in multiple DE stages, the later DE network will produce more accurate estimation, and thus a larger weight should be assigned. Therefore, the geometric sequence with a common ratio $\iota$ (greater than 1) and sum 1 can be used for $\nu_ i$-s, i.e., $\nu_{i}={(\iota-1)}\iota^ {i-1}/{(\iota^{K-1}-1)}$. In addition, the parameter $\alpha_{g}$ is the $g$th element of the indicator vector $\alpha$ for the noise constraint. The reason for introducing $\alpha$ is given in the following. For the commonly used training datasets like SIDD, RENOIR, etc., in real denoising, the specific noise maps $\sigma(\mathbf{x}\_g)$-s are unknown, and thus cannot be used for training the DE network. In this case, the noise map estimation loss should be invalidated by setting its weight to 0; to update the parameters of the DE networks, we need to synthesize the dataset according to the real noise model established in Eq. (2) for training. Therefore, when the training data is synthetic with a known noise map, the weight for the noise map  estimation loss should be 1. This is why we call $\alpha_{g}$ the $g$th element of the indicator vector $\alpha$ for the noise constraint.
>
> We will add these details in the revised version.
>
> >**Mapping noise image to space that better distinguish noise has been proposed in NBNet. Please clarify the difference between you and NBNet on how to use the distinguishable space, and the superiority of your usage.**
>
> \** **The difference between using the distinguishable space:**
>
> + The space projection in NBNet is achieved by the subspace attention (SSA) module, which needs to construct the basis vector for obtaining the projection matrix. It is essentially an attention module. Our method directly using $E$ to project the input image to the distinguishable space without using attention mechanism;
>
> + NBNet uses SSA to project the output of each decoder module separately in a UNet-based architecture, and thus multiple SSA modules are exploited. However, our method only needs to introduce the latent space (LS) at the beginning of the whole network to ensure the whole reconstruction process is carried out in the high-dimensional LS;
>
> + SSA in NBNet requires two inputs, i.e., the low-level feature from skip-connection and the upsampled high-level feature. Specifically, the low-level feature from skip-connection is projected into the signal subspace guided by the upsampled high-level feature. However, our method only needs one single input, i.e., the input noisy image.
>
> \** **Superiority of our usage:**
>
> + To obtain the projection in NBNet, a lot of matrix operations are required. But the projection of our method can be more easily obtained by the $E$ module, avoiding matrix operations;
>
> + NBNet requires two inputs, where one is used as the guidance for the other one. Thus, it cannot be directly used in our work since only one noisy input is available at the beginning of our network. However, since LS only needs a single input, it can directly perform projection on the noisy input image at the beginning of the network.
>
> We will add these analyses in the revised version.
>
> ```
> We hope that the responses alleviate the reviewer's concerns. We are happy to answer any additional questions the reviewer has.
> ```

---

### Official Review · Reviewer_YKat · 2022-07-11

**Rating:** 6
**Confidence:** 3
**Soundness:** 3 good
**Presentation:** 3 good
**Contribution:** 2 fair

**Summary:**

This paper proposes a real-world image denoising model with alternative optimization. The formulation is carefully discussed in order to ensure the validity of the proposed pipeline. This is an interesting approaching to solve the problem of image denoising. Experiments and ablation studies are sufficient to prove the superiority of the proposed method over other state-of-the-art methods and the efficiency of the designed modules.

**Questions:**

1.     The embeddings in the proposed Latent Space (LS) are seemingly just convolutional features of input images. Whether these latent embeddings have special semantics? If not, why the authors emphasize it in the paper (even in the title)?

2.	I notice that the proposed method uses improved channel attention (CIG-CA) and spatial attention (CIG-SA). Have the authors tried self-attention mechanism? If not, why?

3.	The formulation of the proposed method is seemingly weak while the performances are excellent. I totally agree that this is a great work, but I’d like to ask the authors that have you think about to submit your work to ECCV or CVPR? I think it would be more suitable for those conferences instead of NeurIPS.


**Limitations:**

The limitations are clearly addressed in supplementary materials. This work does not have any potential negative societal impact.

**Strengths And Weaknesses:**

Strengths:

1.	The method of alternative optimization is novel to image denoising. In order to adopt this method, the formulation of denoising is strictly discussed and the pipeline is carefully designed.

2.	The definition of the problem and the corresponding solution are clear. Instead of regarding NNs as a ‘black box’, the authors try to analysis the building reasons of every block.

3.	The experiments and ablation studies are sufficient. The performance of the proposed method is better than SotA methods clearly.

Weakness:

1.	The proposed method is seemingly an incremental method to DAN [1]. Although the pipeline is specifically designed for the task of image denoising, I think that it somehow lacks of novelty.

2.	I miss a general description of the total pipeline. Such a complex framework is a little bit hard to read. I suggest the authors to modify the texts of Sec. 2 and 3.

3.	The subjective results shown in the paper is not that impressive. I suggest the author choose clearer results to replace Fig. 4 and the upper row of Fig. 5.

[1] Z. Luo, Y. Huang, S. Li, L. Wang, and T. Tan. Unfolding the alternating optimization for blind super resolution. In Advances in Neural Information Processing Systems (NeurIPS), volume 33, pages 5632–5643, 2020.

---

> ### Author Response · Authors · 2022-08-02
> **Responses to Reviewer YKat**
>
> **Comment:**
> We thank the reviewer for the valuable comments.
>
> >**The proposed method is seemingly an incremental method to DAN. I think that it somehow lacks of novelty.**
>
> Although both DAN and our method adopt alternative optimization (different tasks, i.e., super-resolution (SR) and denoising), **their theoretical novelties and the network novelties are much different.**
>
> \** **Theoretical differences**
>
> + DAN performs SR in low-dimensional pixel space, while ours performs denoising in high-dimensional LS;
>
> + DAN only considers degradation estimation and reconstruction. In addition to degradation estimation ($\mathbf{u}$-Net) and reconstruction ($\mathbf{z}$-Net), we introduce guidance representation (GR, $G$-Module) to improve performance;
>
> + Traditional alternating optimization is used in DAN. But we propose a novel self-correction (SC) alternating optimization method.
>
> \** **Network differences**
>
> + By using SC, our degradation estimation and reconstruction networks can better exploit the last estimates $\mathbf{z}^{(i)}$ and $\mathbf{u}^{(i)}$ for higher performance than DAN that is without SC;
>
> + We introduce $G$-Module to guide denoising, which is not considered in DAN;
>
> + We propose the FM$^2$A module with FSM and DEM, which is different from DAN;
>
> + For $\mathbf{u}$-Net, we propose a novel parameter-free NFAE layer, which is not used in DAN.
>
> We will add the analyses in the ‘Supplementary Material’.
>
> >**I miss a general description of the total pipeline. I suggest the authors to modify the texts of Secs. 2, 3.**
>
> We will add a general description in the revised version. E.g., we will add an introduction at the beginning of Sec. 2; In Sec. 3, we will modify the introduction of each sub-network. The algorithm flow chart corresponding to Sec. 2 has been reported in the ‘Supplementary Material’. Similarly, we will further add a flow chart corresponding to Sec. 3.
>
> >**I suggest the author choose clearer results to replace Fig. 4 and the upper row of Fig. 5.**
>
> We will update these figures to make the visual effect more impressive.
>
> >**Whether these latent embeddings have special semantics? Why the authors emphasize it in the paper?**
>
> Because of the importance of LS, we emphasize it in the paper. LS has the following advantages: 1) better representation ability than the low-dimensional space; 2) allows better information flow in the unfolding network.
>
> Without LS in Eq. (5), denoising is performed in the pixel domain to reconstruct $\mathbf{x}\in\mathbb{R}^{n\cdot c}$. It will lead to the $\mathbf{z}$-Net with a low input/output channel dimension $c$, reducing the information flow in the network.
>
> With LS in Eq. (5), denoising is performed in LS to reconstruct $\mathbf{z}\in{\mathbb{R}^{n\cdot m}}$. Since $m>c$, $\mathbf{z}$-Net has better representation ability. Moreover, the channel number between two consecutive $\mathbf{z}$-Nets becomes larger, and thus the network information flow is enhanced. All of these will benefit the performance.
>
> We can also observe from Sec. 4.5 that LS is important. Moreover, by visualizing the features after $E$, we can find that the embeddings are the hierarchical high-dimensional features of the noisy image, where the noise and image components can be easily decoupled, which can effectively promote the performance.
>
> These reasons and the visual features will be added in the revised version.
>
> >**Have the authors tried self-attention mechanism? If not, why?**
>
> We have considered using self-attention mechanism. However, because of its high parameter number and complexity, we finally adopted FSM for a balance between performance and complexity. Replacing all FSM modules with the self-attention modules in [*1], the parameter number will be 5.3M and the FLOPs will be 421G for a $256\times256$ image, which is much larger than those of our current method. How to use self-attention more efficiently in ScaoedNet will be added as a future work..
>
> [*1] Uformer: A General U-Shaped Transformer for Image Restoration, CVPR 2022
>
> >**The formulation is seemingly weak while the performances are excellent. I totally agree that this is a great work, but have you thought about to submit your work to ECCV or CVPR? I think it would be more suitable.**
>
> ECCV/CVPR indeed are suitable conferences, but we considered the following aspects: 1) our work includes both theoretical novelty and network novelty. The theoretical part is different from the existing methods, and has promising performance; 2) some good works with similar style to ours have been published in NeurIPS, e.g., DAN, [*2, *3]. Therefore, we submitted it to NeurIPS. We will improve the formulation in the revised version.
>
> [*2] Listening to Sounds of Silence for Speech Denoising，NIPS2020.
>
> [*3] Joint Sub-bands Learning with Clique Structures for Wavelet Domain Super-Resolution，NIPS2018.
>
> ```
> We hope that the responses alleviate the reviewer's concerns. We are happy to answer any additional questions that the reviewer has.
> ```

---

> > ### Comment · Reviewer_YKat · 2022-08-07
> > **Further Comment on the Response of Authors**
> >
> > First, I really appreciate your detailed reply to my concerns and questions. Your reply solved most of my concerns.
> >
> > The most important, you differentiated your work from DAN.
> >
> > But on the question of why emphasizing the LS, I still not that persuaded. I can understand that the LS represents better than the low-dimensional space, but to the best of my knowledge, most of the deep-learning based methods of low-level tasks including SR, low-light enhancement, image denoising, image restoration and so on usually use a convolutional layer to transform the input image to its corresponding feature. This seems a convention of the methods of multiple low-level tasks. Can you differentiate your LS from the previous works?
> >
> > At last, based on your reply, I will raise my final rating to BA. But I still expecting your further discussion on the proposed LS.

---

> > > ### Author Response · Authors · 2022-08-09
> > > **Response to Reviewer's Further Concern**
> > >
> > > Thank you for your encouraging response. We are very glad that the reviewer will raise the final rating.
> > >
> > > >**Most of the deep-learning based methods of low-level tasks usually use a convolutional layer to transform the input image to its corresponding feature. Can you differentiate your LS from the previous works?**
> > >
> > > Our paper focuses on the unfolding based real denoising, and thus we first analyze our method and other conventional unfolding based networks for low-level tasks.
> > >
> > > $\circ$$\circ$ **Analyses on our method and other conventional unfolding based networks:**
> > >
> > > The unfolding framework mainly contains two phases: 1) constructing a model-based algorithm for a certain low-level task; 2) designing a network to implement the previous algorithm.
> > >
> > > ①	 ***The drawback of the conventional unfolding based networks:***
> > >
> > > In the conventional unfolding framework, 1) the model-based algorithm is generally based on the traditional maximum a posteriori method, which models the image formation process in image space and integrates image priors into the prediction; 2) the network is designed to implement the iterative steps of the model-based algorithm.
> > >
> > > Since the input and output of the reconstruction update step in the model-based algorithm are in image space (i.e., the algorithm limitation on input and output), the input and output of the corresponding reconstruction sub-network ($\mathbf{z}$-Net) must be in image space as well no matter how $\mathbf{z}$-Net is designed. Therefore, the information flow among (not inside) sub-networks will be constrained to a very low dimension, resulting in performance decrease. **This is a drawback caused by the algorithm limitation in the conventional unfolding framework.**
> > >
> > > ②	***Using LS to break through the limitation of the conventional unfolding framework:***
> > >
> > > LS plays important roles in both the model-based algorithm and network design: 1) for the model-based algorithm, it can model the image formation process in high-dimensional LS and integrate LS priors into the prediction, instead of in the image space; 2) for the network design, the input and output of each $\mathbf{z}$-Net in the unfolding framework are transferred to the high-dimensional LS instead of the image space, which improves the information flow among the sub-networks.
> > >
> > > $\circ$$\circ$ **Differences between our method and other conventional networks for low-level tasks with feature transform on the input image:**
> > >
> > > LS leads to an encoding module $E$ at the beginning of the whole network, which seems to be similar to some conventional networks for low-level tasks, e.g., RCAN [*1] and RDN [*2], that extract features through a convolution layer at the beginning of the whole network. In fact, they have the following differences:
> > >
> > > ①	***Differences in motivation:***
> > >
> > > LS can address the drawback caused by the algorithm limitation in the conventional unfolding framework, and it focuses on the information flow between different reconstruction sub-networks (i.e., $\mathbf{z}$-Nets). In contrast, for the conventional networks extracting feature by a convolutional layer at the beginning of the whole network, the motivation is to achieve the necessary channel number increasing operation in the deep network, instead of considering the information flow among different stages.
> > >
> > > ②	 ***Differences in usage:***
> > >
> > > Some conventional networks for low-level tasks, e.g., RCAN [*1] and RDN [*2], extract features through a convolution layer at the beginning of the whole network, and then use a skip connection from the initial feature to the output feature. Finally, the output feature is mapped to the image space. However, placing a convolution layer at the beginning of the whole network is optional. According to RIDNet [*3] and AINDNet [*4], a skip connection from the input image and output image can be directly used, and a convolution layer can be placed at the beginning of the residual branch instead of the whole network. According to the results reported in various papers, for a certain network, placing a convolution layer at the beginning of the whole network or the residual branch can achieve similar performance.
> > >
> > > In contrast, according to the proposed model-based algorithm, $E$ must be placed at the beginning of the whole network to obtain the LS embeddings. Otherwise, it will reduce to the case of iteration in the image space due to the algorithm limitation on the input and output of $\mathbf{z}$-Net, resulting in significant performance decrease.
> > >
> > > We will add more illustrations about LS in the revised version.
> > >
> > > [*1] Image Super-Resolution Using Very Deep Residual Channel Attention Networks, ECCV 2018
> > >
> > > [*2] Residual Dense Network for Image Super-Resolution, CVPR 2018
> > >
> > > [*3] Real Image Denoising with Feature Attention, ICCV 2019
> > >
> > > [*4] Transfer Learning from Synthetic to Real-noise Denoising with Adaptive Instance Normalization, CVPR 2020

---

> > > > ### Comment · Reviewer_YKat · 2022-08-09
> > > > **Further Comments**
> > > >
> > > > Thank you so much for your detailed discussion of LS!
> > > >
> > > > Your explanation of the reason for using LS is clear enough. You differentiated the encoder $E$ from conventional convolution layers. In my understanding, the main reason of using LS is that the task formulation is based on it. Whether my understanding is correct?
> > > >
> > > > Due to your modeling of the image denoising task and your analysis, I decide to raise my final rating to WA. I suggest the authors to put more analysis of LS in your final version because I'm confused by the reason of emphasizing LS at first (If you have explained it in the paper and I missed it, I'm very sorry).

---

> > > > > ### Author Response · Authors · 2022-08-10
> > > > > **Thanks for Your Kind Comment**
> > > > >
> > > > > Thank you for your useful and kind comment. We appreciate your positive comment about our method, and also appreciate that you raised the final rating. Thank you!
> > > > >
> > > > > For the main reason of using LS, your understanding is correct. It is based on the proposed task formulation, which can break through the limitation of the conventional unfolding framework. We will add more analysis of LS in the final version. Thanks for your suggestion.

---

### Meta-Review · Area_Chair_d1Yx · 2022-08-27

**Recommendation:** Accept
**Confidence:** Certain

**Metareview:**

The paper under review introduces a deep unrolling network driven by a latent space blind model for image denoising. Although the network combines known components, it has novel elements and good algorithms, the experimental results are robust, the implementation details are rich, and the ablation research is extensive. Revisions and rebuttals addressed most of the reviewers' concerns, leading them to improve their scores. Therefore, I accept this paper.

**Award:**

No

---

### Decision · Program_Chairs · 2022-09-14

Accept